# Capsaicin Inhibits Inflammation and Gastric Damage during *H pylori* Infection by Targeting NF-kB–miRNA Axis

**DOI:** 10.3390/pathogens11060641

**Published:** 2022-06-01

**Authors:** Kalyani Saha, Deotima Sarkar, Uzma Khan, Bipul Chandra Karmakar, Sangita Paul, Asish K. Mukhopadhyay, Shanta Dutta, Sushmita Bhattacharya

**Affiliations:** 1Department of Biochemistry, National Institute of Cholera and Enteric Diseases, Indian Council of Medical Research (ICMR-NICED), P-33, CIT Rd, Subhas Sarobar Park, Phool Bagan, Beleghata, Kolkata 700010, India; kalyan.saha86@gmail.com (K.S.); deotima_s@yahoo.com (D.S.); uzma.khan1493@gmail.com (U.K.); 2Department of Microbiology, National Institute of Cholera and Enteric Diseases (ICMR-NICED), Indian Council of Medical Research, P-33, CIT Rd, Subhas Sarobar Park, Phool Bagan, Beleghata, Kolkata 700010, India; bkniced032@gmail.com (B.C.K.); sangitapaul13292@gmail.com (S.P.); mukhopadhyayak.niced@gov.in (A.K.M.); 3Department of Bacteriology, National Institute of Cholera and Enteric Diseases, Indian Council of Medical Research (ICMR-NICED), P-33, CIT Rd, Subhas Sarobar Park, Phool Bagan, Beleghata, Kolkata 700010, India; shanta.niced@icmr.gov.in

**Keywords:** *H.* *pylori*, capsaicin, miRNA, NF-kB, cytokines, inflammation

## Abstract

*Helicobacter pylori (H. pylori)* infection is considered as one of the strongest risk factors for gastric disorders. Infection triggers several host pathways to elicit inflammation, which further proceeds towards gastric complications. The NF-kB pathway plays a central role in the upregulation of the pro-inflammatory cytokines during infection. It also regulates the transcriptional network of several inflammatory cytokine genes. Hence, targeting NF-kB could be an important strategy to reduce pathogenesis. Moreover, treatment of *H. pylori* needs attention as current therapeutics lack efficacy due to antibiotic resistance, highlighting the need for alternative therapeutic approaches. In this study, we investigated the effects of capsaicin, a known NF-kB inhibitor in reducing inflammation and gastric complications during *H. pylori* infection. We observed that capsaicin reduced NF-kB activation and upregulation of cytokine genes in an in vivo mice model. Moreover, it affected NF-kB–miRNA interplay to repress inflammation and gastric damages. Capsaicin reduced the expression level of mir21 and mir223 along with the pro-inflammatory cytokines. The repression of miRNA further affected downstream targets such as e-cadherin and Akt. Our data represent the first evidence that treatment with capsaicin inhibits inflammation and induces antimicrobial activity during *H. pylori* infection. This alternative approach might open a new avenue in treating *H. pylori* infection, thus reducing gastric problems.

## 1. Introduction

Gastric cancer is one of the leading causes of mortality worldwide. *H. pylori*, a common gastric pathogen, is considered to be one of the most important carcinogens involved in gastric cancer [1,2,3,4,5,6]. Although only 2% of infected individuals develop cancer, it also causes other gastric complications such as ulcer and gastritis. In the beginning, infection leads towards inflammation and gastric damage, but in later stages, tumor development occurs [6]. The important paradigm is about the progression of the disease—how *H. pylori* infection gradually migrates towards gastric complications. There are several factors involved during *H. pylori* pathogenesis—genetics, host environment, and virulence factors [7,8,9]. During the infection process, the pro-inflammatory cytokine levels rise. This cytokine upregulation is triggered by virulence factors secreted by *H. pylori* [7,8,10,11]. One of the factors, CagA, plays an important role in inflammatory responses. CagA secreted by *H. pylori* is responsible for activating critical pathways such as NF-kB, Akt, src/MEK/ERK [7,12]. In fact, the virulence factors can trigger different gene expression cascades, resulting in activation of pro-inflammatory transcription factors such as NF-kB, AP1, and production of cytokines to facilitate gastric damage. NF-kB plays a major role in controlling the inflammatory cytokine signaling pathway [8,13,14]. It binds to the promoter of different cytokines and regulates their gene expression [12,15,16,17,18]. Cytokines such as IL-6, when upregulated, further activate transcription factors to maintain an inflammatory environment [19]. NF-kB, on the other hand, cooperates with other transcription factors to induce inflammation. These transcription factors bind not only to the promoter of cytokine genes, but also to the promoter sites of the micro RNAs [10,20,21,22,23]. The transcription factor miRNA network has been associated with the regulation of pro-inflammatory responses that further enhance gastric problems during *H. pylori* infection [21,22]. Among several NF-kB regulated miRNAs, miR21 and miR223 play a key role in pathogenesis [10,23]. In addition, CagA, a virulence factor released by *H. pylori*, is responsible for an increase in NF-kB mediated transcription of miRNAs such as miR223 [10]. NF-kB has binding sites localized in miR21, which is known to be upregulated during gastric disorders [21]. Moreover, the miRNA–NF-kB network is responsible for interlinking inflammation with gastric complications. Hence, targeting these miRNAs may serve as a novel approach for intervention in *H. pylori* infection.

*H. pylori* infection is commonly treated with antibiotics and proton pump inhibitors [24,25]. However, antibiotic resistance is a major concern responsible for the failure of eradication therapy, and thus, failure often leads to development of carcinogenesis [26,27]. Hence, alternative approaches are needed for intervention.

Capsaicin is a bioactive compound isolated from the plant *Capsicum chinense* [28,29]. It is a potent anticancer, anti-inflammatory, antioxidant and antidiabetic agent [30,31]. Capsaicin is commonly used as a topical analgesic and is found in many formulations of creams [32]. It is also used in some dietary supplements [33]. It is known to inhibit inflammation during *H. pylori* infection in in vitro conditions; however, its role in inflammation and miRNA expression has not been explored in in vivo conditions [34].

The current study focused on the anti-inflammatory effect of capsaicin in in vitro and in vivo conditions during *H. pylori* infection by targeting the NF-kB–miRNA inflammatory axis. As inflammation induces gastric damages, we further assessed the role of capsaicin in gastric damage.

## 2. Results

### 2.1. Capsaicin Reduces Cytokine Production in In Vitro and In Vivo Conditions

Previous reports showed that capsaicin inhibits NF-kB mediated IL-8 production in *H. pylori* infected gastric cancer cell lines [28]. Moreover, it is known to be effective as an immune potentiator [35]. Hence, we assessed the anti-inflammatory effect of capsaicin during *H. pylori* infection in both in vitro and in vivo conditions. Here, for the first time, we observed that capsaicin reduces the level of other cytokines IL-6 and TNFα in both in vitro and in vivo conditions. Capsaicin-pretreated gastric cells (100 μM) were infected with *H. pylori* (SS1 strain, a CagA positive isolate) at 1:100 MOI for 4 h. Results showed that capsaicin decreased the elevated level of cytokines significantly in *H. pylori* infected cells as compared to untreated cells We also checked cytotoxicity of capsaicin in AGS cells. A 100 μM concentration of capsaicin showed insignificant toxicity (Appendix A). This is consistent with the previous literature [36] (Figure 1A). Furthermore, to assess the effect of capsaicin in an in vivo model of *H. pylori*, we infected mice with *H. pylori* SS1 strain [19,37]. After infection, mice were treated with capsaicin for 40 days at 5 mg/kg body weight dose. Capsaicin reduced cytokine (IL-6 and TNFα) production in the treated group significantly as compared to the *H. pylori* infected, non-treated group (Figure 1B). Taken together these results suggest that capsaicin reduces the level of pro-inflammatory cytokines in both in vitro and in vivo models of *H. pylori* infection.

### 2.2. Effect of Capsaicin on Inflammatory Gene Expression

Several factors are involved in regulating the expression of the pro-inflammatory cytokines. Therefore, in the next step, we checked the expression of cytokines at the gene level. We measured the transcript levels of IL-6, TNFα and IL-1β (Figure 2A–D). The mRNA levels of all the cytokine genes (IL-6, TNFα, and IL-1β) were overexpressed due to *H. pylori* infection in mice gastric tissues. Further, we checked NLRP3 as it is involved in inflammation and production of IL-1β [18,38]. Capsaicin repressed mRNA expression of NLRP3 and cytokines (IL-6, TNFα, and IL-1β) significantly in infected tissues as compared to untreated samples. As the transcription factor NF-kB is known to be responsible for elevated expression of NLRP3 inflammasome and the cytokines, we examined the activation status of NF-kB due to drug exposure [38]. The phosphorylation status of NF-kB in mice tissue samples was analyzed by western blot. Consistent with previous reports in in vitro systems, we observed that capsaicin exposure reduced NF-kB phosphorylation (Figure 2E). Hence, these results indicate that capsaicin treatment reduced inflammation during *H. pylori* infection in a mouse model by targeting NF-kB activation.

### 2.3. Capsaicin Inhibits NF-kB Regulated miRNA Gene Expression and H. pylori Infection

It is reported that during *H. pylori* infection, miRNAs such as miR21 and miR223 are upregulated. NF-kB is known to bind to the promoter sites of miR21 and miR223 [4,10,21]. Hence, we checked the targets for NF-kB in *H. pylori* infected mice tissues. Capsaicin exposure reduced the expression of miR21 and miR223 significantly as compared to *H. pylori* infection (Figure 3A,B). In addition, we checked *H. pylori* infection levels in mice tissues by real-time PCR (16S rDNA). Capsaicin treatment for 40 days at 5 mg/kg body weight reduced *H. pylori* infection significantly in gastric tissue samples as compared to infected mice (Figure 3C). Furthermore, we investigated the effect of capsaicin in NF-kB promoter activity. Briefly, AGS cells were pretreated with capsaicin (100 μM), followed by *H. pylori* infection for 4 h. Nuclear extracts were examined for determining transcriptional activity of NF-kB. *H. pylori* infection stimulated binding of NF-kB to NF-kB response elements. In contrast, capsaicin significantly attenuated *H. pylori* induced NF-kB DNA binding activity. Additionally, we checked the antimicrobial action of capsaicin in in vitro cellular conditions. We pretreated the cells with capsaicin (100 μM) and then infected them with *H. pylori* for 4 h. As CagA is involved in inflammation and colonization of *H. pylori*, we checked the virulence factor CagA expression by Western blot. CagA expression decreased due to capsaicin treatment in *H. pylori* infected cells (Figure 3D). Collectively, these results suggest that capsaicin inhibits NF-kB-dependent miRNA upregulation and also inhibits *H. pylori* infection.

### 2.4. Capsaicin Exposure Affects Downstream Targets of miRNA 21 and 223

To elucidate the effect of capsaicin on downstream targets of the miRNAs, we examined the expression of e-cadherin and Akt phosphorylation. Both miR21 and miR223 are known to be involved in epithelial–mesenchymal transition (EMT) and Akt activation [37,38]. The mir21 and mir223–NF-KB axis are reported to downregulate e-cadherin [10,38,39]. In addition, it has been reported that e-cadherin downregulation takes place and that Akt phosphorylation is increased during infection [40,41,42]. Hence, we checked e-cadherin expression and Akt phosphorylation by immunoblotting. Capsaicin upregulated e-cadherin expression significantly in *H. pylori* infected mice tissues, whereas Akt phosphorylation was inhibited by capsaicin (Figure 4A,B). All of these findings indicate that capsaicin downregulates Akt activation and overexpresses e-cadherin expression in *H. pylori* infected mice tissues.

### 2.5. Capsaicin Treatment Reduces Gastric Tissue Damage

To further assess the anti-*H. pylori* effect of capsaicin, we examined gastric tissues for changes in morphology. There were changes in the gastric histopathology when compared with control group for *H. pylori* infected gastric tissues (Figure 5A). *H. pylori* infection increased inflammation and inflammatory cell infiltration in gastric tissues, which resulted in epithelial cell damage. However, capsaicin, on the other hand, decreased inflammation in infected tissues (Figure 5B). Finally, we performed in silico analysis to determine how capsaicin is able to inhibit NF-kB activity. Results showed that capsaicin binds to the promoter site of NF-kB. Table 1 shows binding efficiency of capsaicin with NF-kB promoter element, which is based on the analysis of hydrogen bond interaction, hydrophobic interaction, and van der Waals interactions (Figure 5C). These results summarize that capsaicin is capable of reducing inflammation and gastric cancer prognosis by inhibiting NF-kB activation.

## 3. Discussion

Chronic or acute infection by pathogens is a severe threat to humans. *H. pylori* is one of the pathogens involved in gastric disorders [1,2,3,4,5,6]. Infection by *H. pylori* not only induces gastric damage but in severe cases promotes gastric cancer [43]. This Gram-negative bacterium modulates the cellular milieu by interfering with host gene and protein expressions, making the host niche suitable for gastric disorders [6]. Reprogramming of the signaling pathways by *H. pylori* promotes gastric complications such as tissue damage, ulcers, and in severe cases, cancer. Several virulent factors secreted by *H. pylori* induce inflammation. This inflammation further proceeds towards gastric complications such as peptic ulcer, gastritis and cancer [7,8,9]. One of the most important virulent factors is CagA, secreted by *H. pylori.* CagA is associated with inflammation. The triggering of inflammation is due to activation of different host proteins by the virulent factors [10,11]. NF-kB, a transcription factor, is activated due to infection and plays a central role in upregulation of pro-inflammatory cytokines [12,13,14].

The current study demonstrates that the herbal compound capsaicin significantly reduced inflammation during *H. pylori* infection by inhibiting NF-kB activation. Previously, it has been reported that capsaicin reduces inflammation in *H. pylori* infected gastric cancer cells by targeting NF-κB activation [28]. However, the role of capsaicin in inhibiting inflammation in in vivo models is unknown. Therefore, the main aim of the current study was to determine the ability of capsaicin to reduce NF-kB activation in an in vivo model. Capsaicin reduced NF-kB phosphorylation, which was upregulated due to infection. This inhibition of NF-kB affected the downstream targets of inflammation. Inflammatory markers IL-6 and TNFα were reduced due to capsaicin treatment in both in vitro and in vivo conditions. Besides the expression of inflammatory markers at the protein level, we examined the expression levels of these markers at the gene level. Coincidently, gene expression analysis showed that the inflammatory genes IL-6, TNFα and IL-1β were downregulated due to capsaicin treatment. NF-kB binds to the promoter of these inflammatory cytokines and NLRP3 [15,16,17,18]. NLRP3 is known to activate IL-1β production, and it is normally upregulated during *H. pylori* infection [44]. Here, we investigated NLRP3 as NF-kB activates NLRP3 gene transcription. Probably, inactivation of NF-kB by capsaicin inhibited the expression of the cytokines and NLRP3 genes.

It is known that NF-kB-dependent activity is a pivotal link between inflammation and gastric disorders. However, the role of capsaicin in NF-kB inactivation and its downstream effects during *H. pylori* infection are unknown. NF-kB targets several miRNAs. NF-kB binds to the promoter of the miRNAs that induce gastric problems [19,20,21,22]. mir21 and miR223 belong to the class of NF-kB-dependent miRNAs that are reported to be upregulated in gastric cancer [10,23]. Additionally, they are overexpressed during *H. pylori* infection. Herein, capsaicin repressed gene expression of miR21 and mir223. Since NF-kB binds to the promoter of cytokines and miRNAs, we examined the effect of capsaicin on NF-kB promoter activity. Consistently, capsaicin suppressed *H. pylori* induced NF-kB transcriptional activity. Subsequently, we studied the targets of the miRNAs. We checked a transmembrane glycoprotein molecule known as e-cadherin, which plays a significant role in sustaining cell–cell adhesion and is found to be downregulated during *H. pylori* infection [23,45]. Moreover, mir21 and miR223 are collectively known to downregulate e-cadherin expression [46,47]. In this study, we demonstrated that capsaicin upregulated e-cadherin expression in *H. pylori* infected mice tissues. Thus, capsaicin targets the NF-kB–miRNA axis to inhibit *H. pylori* infection.

Further we examined Akt phosphorylation as Akt activation is known to be induced by miR21 and mir223. Phosphorylation of Akt and NF-kB are known to be directly linked with inflammation and gastric damage. Capsaicin reduced Akt phosphorylation in infected mice tissues, confirming the anti-inflammatory effect. As capsaicin inhibited inflammation, we further assessed the anti *H. pylori* effect. Capsaicin reduced *H. pylori* count in gastric tissues and reduced the level of CagA expression in the examined cell line. Therefore, capsaicin was able to restrict inflammation and at the same time also reduce infection. Further studies are required to understand whether there is any direct effect of capsaicin on CagA expression.

Finally, capsaicin was able to inhibit gastric tissue damage in an in vivo mice model. Histological studies revealed the therapeutic effect of capsaicin. Capsaicin reduced inflammation and gastric tissue damage in *H. pylori* infected tissues. As capsaicin reduced inflammatory markers at the gene level, the resultant effects were observed in gastric tissue morphology. Further, to understand the basis of NF-kB inactivation by capsaicin, we performed in silico studies. Previously, it has been shown that capsaicin reduced NF-kB promoter activity in cell lines [28]. Hence, we performed in silico analysis and showed that capsaicin has the potential to bind to the promoter site of NF-kB. Nevertheless, in future, more investigations are required to understand direct promoter site binding by capsaicin.

In this study, we hypothesized and showed that the anti-inflammatory property of capsaicin can be exploited for improvement of *H. pylori* infection. Capsaicin reduced NF-kB activation and thereby intervened in the inflammatory link with pathogenesis. Here, we showed that NF-kB-regulated transcription of inflammatory cytokines and miRNAs were decreased by capsaicin. Moreover, we demonstrated in silico binding of capsaicin to NF-kB promoter, but further investigations are needed to confirm direct promoter binding ability of capsaicin to the NF-kB promoter site. Capsaicin showed antimicrobial activity against *H. pylori* along with its anti-inflammatory effects. Finally, capsaicin is a pharmacologically novel therapeutic in inhibiting inflammation that probably can reduce gastric problems during *H. pylori* infection.

## 4. Materials and Methods

### 4.1. Chemicals and Reagents

Chemicals and reagents used were purchased from Sigma (St. Louis, MO, USA), Santacruz (Dallas, TX, USA), Cell Signaling Technology (Danvers, MA, USA), and Abcam (Cambridge, UK). Capsaicin was purchased from Sigma Aldrich (CAS Number: 404-86-4).

### 4.2. H. pylori Culture

The mouse colonizing *H*. *pylori* strain SS1 (*cagA*^+^, *vacA s2m2*) was used for this study. Frozen stocks of *H. pylori* strains were revived on brain heart infusion (BHI) agar (Difco Laboratories, Detroit, MI, USA) supplemented with 7% heat-inactivated horse serum (Invitrogen, NY, USA), trimethoprim (5 mg/L), vancomycin (8 mg/L), polymyxin B (10 mg/L) and 0.4% IsoVitaleX (Becton Dickinson, Sparks, MD, USA). The plates were incubated at 37 °C in a microaerophilic atmosphere (5% O_2_, 10% CO_2_, 85% N_2_) (double gas incubator) for 3 to 6 days. Stock cultures were maintained until use at −70 °C. Isolates were re-streaked on fresh BHI agar and incubated for 24 h for use in downstream work.

### 4.3. Cell Culture

AGS cells (ATCC CRL 1739, a human gastric adenocarcinoma cell line) were grown in F12 (GIBCO BRL, Germany) with 10% heat-inactivated FBS and 1% pen-strep (penicillin 100 units/mL and streptomycin 100 µg/mL) in an incubator at 37 °C and 5% CO_2_.

### 4.4. Co-Culture Assay

Cells were seeded at a density of 0.5 × 10^6^ cells/mL per well into 6-well costar tissue culture plates. At this seeding density, 80% confluency of monolayers was obtained at the time of the experiment. *H. pylori* culture was suspended in sterile phosphate buffered saline (PBS) and adjusted to an optical density (OD) of 1 at 600 nm, followed by centrifugation at 10,000× *g* for 10 min. The cells were washed with PBS (pH 7.4) and incubated in antibiotic and FBS- free incomplete (F12) media for 2 h. For drug treatment, cells were pre-treated with capsaicin (100 μM) for 2 h, followed by infection with multiplicity of infection of bacteria (MOI 100) for 4 h. The cells and media supernatant of each well were collected after infection and centrifuged at 6000× *g* for 10 min at 4 °C.

### 4.5. Mice

C57BL/6 wild-type mice bred in-house were maintained ad libitum in 24 h dark light cycles in our animal house. Mice aged 8 weeks were selected for the experimental procedures. The animal experiments were performed following all guidelines of the ICMR-NICED Animal Research Ethics (PRO 155).

#### *H. pylori* (ATCC SS1) Infection in C57BL/6 Mice

Eight-week-old pathogen-free male mice were acclimatized for 14 days. After acclimatization, the mice were treated every day for 7 days with antibiotic cocktail (Ciprofloxacin, Metronidazole, Erythromycin, Albendazole). Further, 12 mice were orally gavaged with *H. pylori* (0.2 mL) SS1 strain, whereas 6 control mice housed in a separate cage, i.e., control group, were treated with saline. Mice received 10^8^ CFUs of *H. pylori* by gastric intra-gavage every alternate day for a week [48,49] and were kept quarantined for 14 days to generate gastric damage. After a quarantine period of 14 days, mice in the *H. pylori* inoculated group were treated with capsaicin (5 mg/kg body weight) for 40 days, i.e., the *H. pylori* + Capsaicin treated group, and other 6 mice were kept without capsaicin treatment, i.e., the *H. pylori* treated group. The dose of capsaicin was selected in the infection model based on available data on capsaicin [50]. The treatment procedure is represented diagrammatically in Figure 1C. At the end of the treatment regime, all mice were sacrificed (Control, *H. pylori* treated, and *H. pylori* + Capsaicin treated groups), and gastric tissues and blood were collected for experimental purposes.

### 4.6. MTT Assay

AGS cells were seeded at a density of 0.5 × 10^6^ cells/mL per well into 6-well tissue culture plates, and different doses of capsaicin (50 μM, 100 μM, 200 μM and 250 μM) were added. After 24 h of incubation, 20 μL of MTT was added and the wells were incubated for 5 h. A 300 μL amount of DMSO was used to stop the colorimetric reaction, and the O.D. was measured at 570 nm. Percentage viability of cells was compared with that of control and represented graphically.

### 4.7. Enzyme-Linked Immunosorbent Assay (ELISA) of Cytokine

Pro-inflammatory cytokine (TNF-α, IL-1β, IL-6) levels from media and serum were estimated using BD Biosciences (Franklin Lakes, NJ, USA) cytokine ELISA kit and Krishgen Biosystems (Maharashtra, India) kit as per manufacturer’s instructions. All experiments were performed in triplicate.

### 4.8. Western Blot Analysis

Total protein extracts were prepared from gastric tissues and AGS cells. Cellular and tissue proteins were resolved in 10% SDS–PAGE gels, transferred to polyvinylidene fluoride (PVDF) membranes, and subsequently blocked in 5% BSA in Tris-buffered saline with Tween 20 (TBST). Membranes were then incubated with specific primary antibody for overnight at 4 °C. After washing, membranes were further incubated with secondary antibodies at room temperature for 2 h. The membranes were washed with TBST, and bands were detected using the Chemi-Doc Imaging system. The levels of target protein were normalized with a housekeeping control. All experiments were performed in triplicates. Antibodies utilized are listed in the Appendix A.

### 4.9. Real-Time PCR

RNA isolation was performed using RNA isolation kit (Qiagen research) from AGS cells. Total RNA was extracted using the TRIzol method for mice tissues. cDNA synthesis kit (Thermo Scientific) was used to convert isolated RNA into cDNA. qRT PCR was performed using SYBR green with different mouse primer sequences of cytokine genes (Appendix A).

For miRNA detection, cDNA synthesis was performed using the stem-loop refolding method. Briefly, stem-loop primer stock was overlayed with molecular biology grade mineral oil; it was then heated to 95 °C for 10 min, cooled to 75 °C, and held at 68 °C, 65 °C, and 62 °C for 1 h each. Finally, before transfer from under oil, primers were held at 60 °C for several hours. Modified PCR protocols and specific stem-loop primers with universal reverse primer were used for cDNA synthesis of each sample (primer sequences are described in Appendix A). The expression pattern and Ct values were observed quantitatively from real-time PCR, and relative changes in gene expressions were calculated with the 2^−∆∆Ct^ method using internal controls. Fold change was plotted in a column graph to observe differences amongst different groups.

### 4.10. NF-kB Promoter Assay

The NF-kB transcription factor assay was performed using NF-KB transcription factor assay kit (Abcam—ab133112) following the manufacturer’s protocol. Nuclear extracts were prepared from control, *H. pylori* infected, *H. pylori* infected + capsaicin-treated and only-capsaicin-treated AGS cells. Samples were added to wells coated with oligonucleotides containing the kB promoter sequence. After the addition of primary antibody, HRP-conjugated secondary antibody was added. This step was followed by the addition of TMB substrate reagent, and the reaction was stopped with stop solution, and O.D. collected at 450 nm by spectrophotometer. The fold change was calculated and graphically represented.

### 4.11. Histopathological Study

Gastric tissues containing antrum and corpus regions were collected from the respective groups after sacrifice, and placed in 10% buffered formalin for tissue fixation. Dehydration was performed by graded ethanol (50–100%) treatment followed by xylene. Tissues were then embedded in paraffin (56–58 °C) at 58 ± 1 °C for 4 h. Deparaffinization of the paraffin sections was performed done with xylene. The slides were stained with hematoxylin-eosin and mounted in DPX under a clean cover slip. A bright field microscope was utilized to observe histological changes (Motic, Germany), and photographs were captured at different magnifications.

### 4.12. In Silico Studies

For preparation of protein and ligand structures the three-dimensional structures of capsaicin and NF-kB p65 were downloaded from RCSB Protein Data Bank server. After downloading, UCSF Chimera version 1.11 was utilized to visualize all of the protein structures [51]. The protein was prepared for the purpose of docking by addition of hydrogens to the structure utilizing UCSF Chimera and then saved as a new PDB file.

The 3D structure of the ligand (capsaicin) in SDF format was obtained from PubChem (Compound CID: 1548943) and converted to a PDB file using UCSF Chimera [50].

For molecular docking studies, the Patchdock and Firedock online docking servers were utilized. The Patchdock server is based on a rigid body docking algorithm based on the principles of geometrical shape complementarity. BIOVIA Discovery Studio was used for visualization of the docked complexes (Sharma et al., 2021) [52].

### 4.13. Statistical Analysis

Statistical analysis was performed in Graphpad Prism 5. Data were analyzed using one-way ANOVA (n = 3) followed by Tukey’s post hoc test. Values have been expressed as mean ± standard error of mean (SEM).

## Figures and Tables

**Figure 1 pathogens-11-00641-f001:**
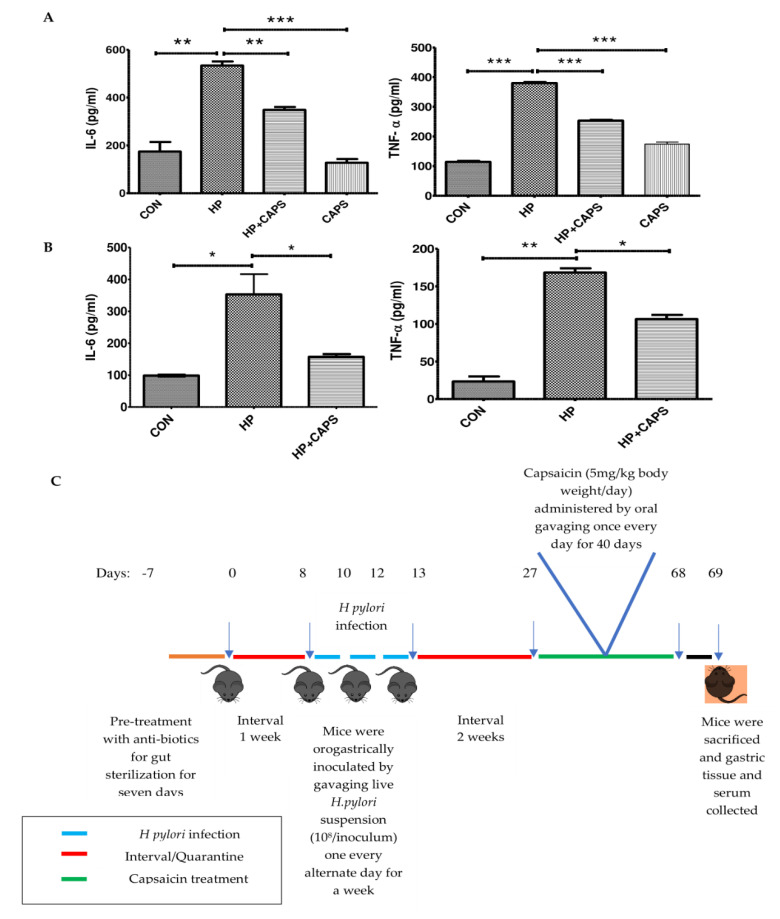
Capsaicin ameliorates *H. pylori* induced inflammation in both in vitro and in vivo conditions. (**A**) ELISA was performed with the media collected from Control (**CON**), *H. pylori* infected (**HP**) and *H. pylori* infected + Capsaicin treated (**HP + Caps**) and Capsaicin (Caps) treated samples to check the levels of cytokines IL-6 and TNFα. AGS gastric cells were pretreated with capsaicin (100 μM) for 2 h and then infected with *H. pylori* (MOI 100) for 4 h. (**B**) Serum samples collected from control (**CON**), infected (**HP**) and infected mice treated with capsaicin (**HP + Caps**) were subjected to ELISA, showing the effect of capsaicin treatment in the levels of cytokines IL-6 and TNFα in *H. pylori* infected mice. Data are represented as mean ± SEM. Values as tested by ANOVA followed by Tukey’s post hoc test. ‘*’, ‘**’, and ‘***’ represent significant differences between groups at *p* < 0.05, *p* < 0.01, and *p* < 0.001, respectively. Experiments were performed in triplicates. (**C**) Mice were treated with antibiotic cocktail every day for 7 days. After an interval of 1 week, mice were infected with *H. pylori* by gastric intra-gavage every alternate day for a week. After 14 days of quarantine period, mice in the *H. pylori* inoculated group were treated with capsaicin (5 mg/kg body weight) for 40 days as the *H. pylori* + capsaicin-treated group. At the end of the treatment regime, all mice were sacrificed, and gastric tissues and serum were collected.

**Figure 2 pathogens-11-00641-f002:**
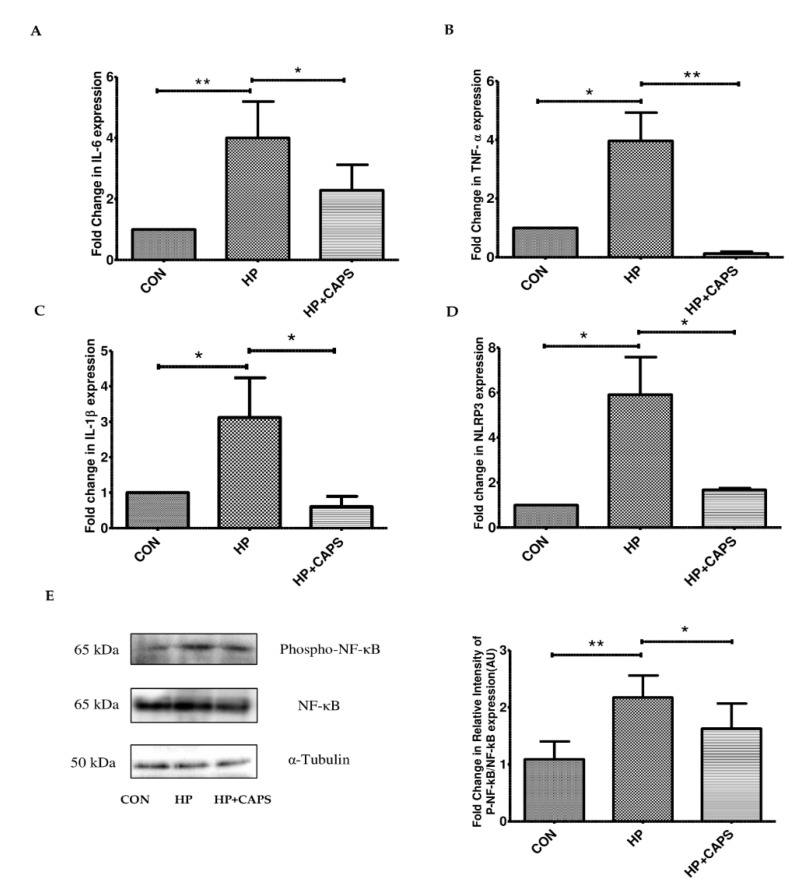
**Treatment with capsaicin affects the expression of NF-kB regulated genes.** (**A**–**D**) Capsaicin treatment decreased the mRNA expression levels of IL6, TNFα, IL-1β and NLRP3 as compared to infected tissues. Values were normalized to GAPDH as housekeeping gene for indicated genes. Control **(CON)**, *H. pylori* infected **(HP)**, *H. pylori* infected + Capsaicin treated **(HP + Caps)**. (**E**) Immunoblot showing the expression of phospho NF-κB and NF-κB after capsaicin treatment in *H. pylori* infected mice. Experiments were performed in triplicates. α Tubulin was used as internal control. Densitometry of representative blot was performed. Data are represented as mean ± SEM. Values were tested by ANOVA followed by Tukey’s post hoc test. ‘*’and ‘**’ represent significant differences between groups at *p* < 0.05, *p* < 0.01, respectively.

**Figure 3 pathogens-11-00641-f003:**
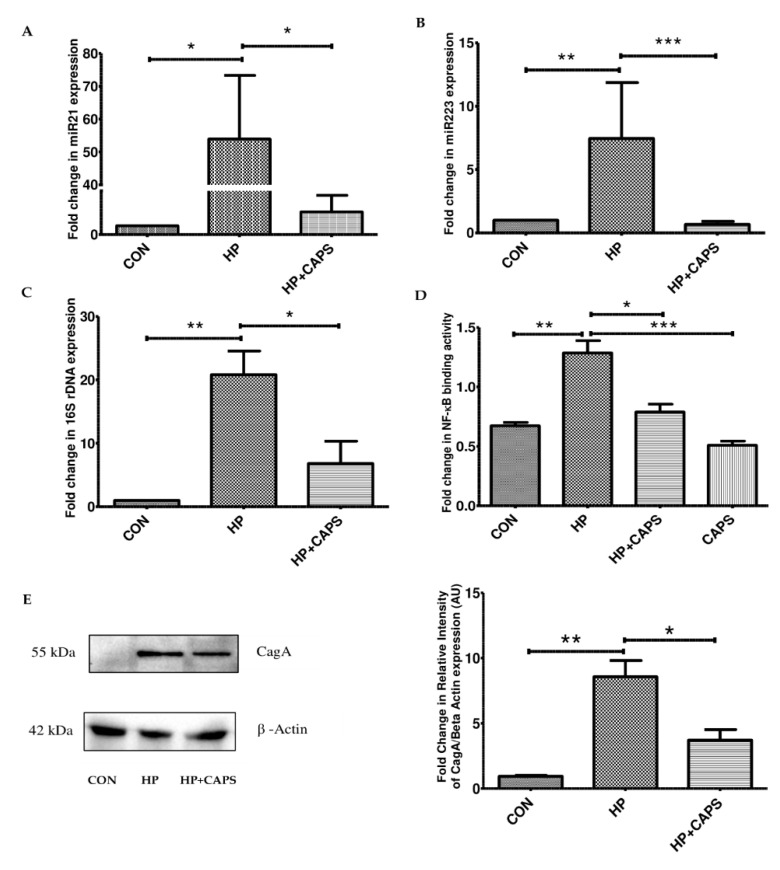
**Capsaicin reduces NF-kB-regulated miRNA expression and *H. pylori* infection.** (**A**,**B**) Real-time PCR showing the expression levels of miR21 and mir223 in capsaicin-treated mice samples as compared to *H. pylori* infected samples. In case of miRNAs, U6 was used as control to normalize miRNAs. Control (**CON**), *H. pylori* infected (**HP**), *H. pylori* infected + Capsaicin treated (**HP + Caps**). (**C**) Real-time PCR was performed to detect *H. pylori* specific 16s ribosomal DNA. (**D**) Control (**CON**), *H. pylori* infected (**HP**), *H. pylori* infected + Capsaicin treated (**HP + Caps**) and only Capsaicin (CAPS) treated AGS cells were subjected to NF-kB promoter assay. Promoter assay was performed to show the activation of NF-kB binding to the promoter sequence. (**E**) Western blot showing *H. pylori* infection increased expression of virulence factor CagA, while capsaicin treatment (100 μM) decreased it significantly in AGS cell line. β actin was used as loading control. Densitometry of representative blot was performed, and values are tested by ANOVA followed by Tukey’s post hoc test. ‘*’, ‘**’ and ‘***’ represent significant differences between groups at *p* < 0.05, *p* < 0.01, *p* < 0.001, respectively.

**Figure 4 pathogens-11-00641-f004:**
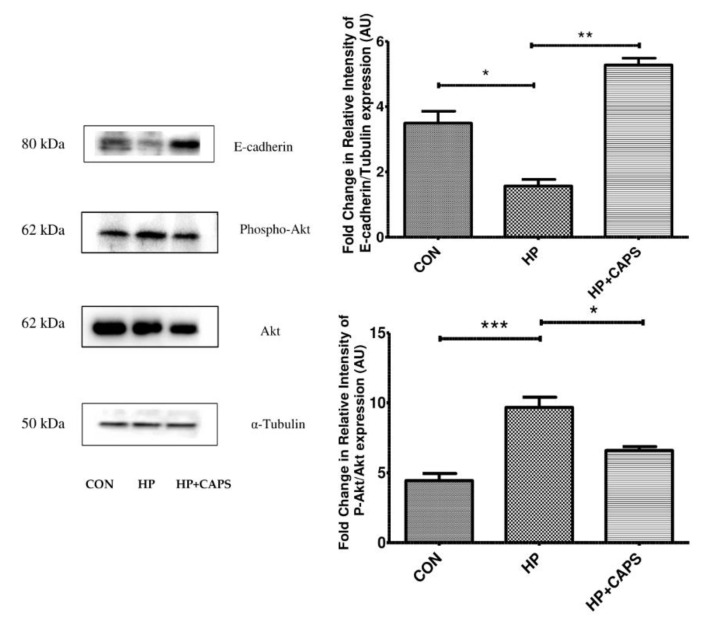
**Capsaicin exposure affects e-cadherin and pAkt expression.** E-cadherin, phospho Akt and Akt levels were determined by Western blot in control **(CON)**, *H. pylori* infected **(HP)** and capsaicin treated gastric tissue samples **(HP + Caps)**. Densitometric analysis was performed keeping total Akt and α tubulin as the internal control. Data are represented as mean ± SEM. Values as tested by ANOVA followed by Tukey’s post hoc test. ‘*’, ‘**’, and ‘***’; represent a significant difference between groups at < 0.05, *p* < 0.01, *p* < 0.001, respectively.

**Figure 5 pathogens-11-00641-f005:**
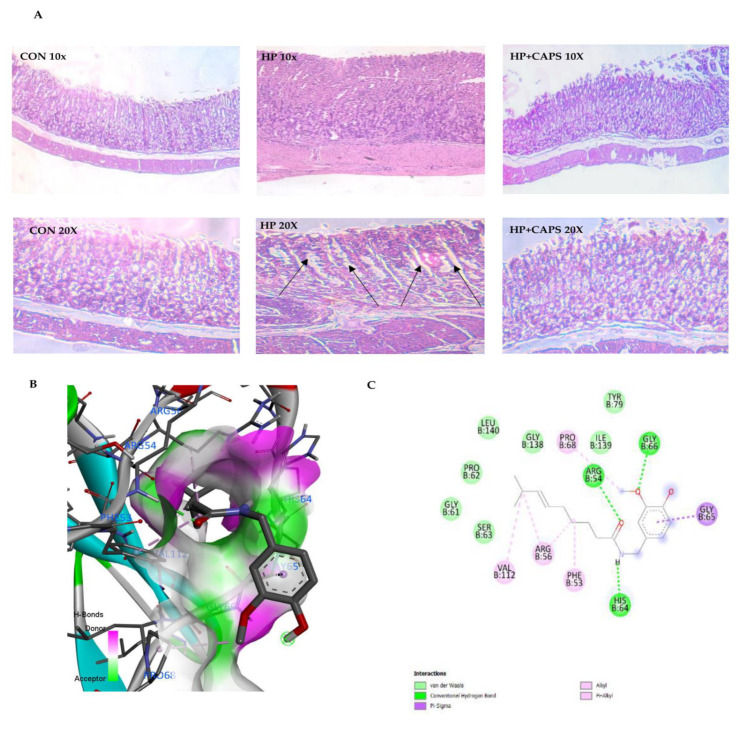
**Capsaicin reduced gastric tissue damage.** (**A**) *H. pylori* infection causes overall inflammation along with distortion of gastric tissue. Inflammatory changes are represented by black arrows. Capsaicin treatment ameliorates the effects of *H. pylori* infection significantly. Control gastric tissue at 10× and 20× (**left panel-CON**). *H. pylori* treated gastric tissue at 10× and 20× (**middle panel-HP**) and capsaicin-treated infected gastric tissue (**Right panel-HP + Caps**) after *H. pylori* infection in mice. (**B**) Capsaicin binds strongly to NF-κB promoter binding domain. 3D representation of docking complexes of NF-κB with capsaicin as visualized in BIOVIA discovery studio after docking. Docking was performed using Patchdock and Firedock web servers. (**C**) 2D representation of docked complexes to show hydrogen bonds and amino acid interactions.

**Table 1 pathogens-11-00641-t001:** In silico studies on interaction between NF-kB promoter binding motif and capsaicin.

Receptor	Ligand	Binding Energy (Kcal/mol)	H-Bond Interactions	Hydrophobic Interactions	VDW Interactions	PDB ID
NF-kB promoter binding motif	Capsaicin	−5.82	GLY66, ARG54, HIS64	GLY65, PRO68, PHE53, ARG56, VAL112	SER63, GLY61, PRO62, LEU140, GLY138, ILE139, TYR79	2061

## Data Availability

All data are included in the article and Appendix A.

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
