# Peer review of "Capsaicin Inhibits Inflammation and Gastric Damage during H pylori Infection by Targeting NF-kB–miRNA Axis"

_pathogens, 2022, doi:10.3390/pathogens11060641_

Round 1

Reviewer 1 Report

Major comments:

  • Lines 74 to 76: Authors need reference(s) for their statement that Capsaicin is known to inhibit H pylori infection.

  • Figure 1: Is this a typo or are the authors using 100M concentration of the drug? Please provide cell viability assay for that drug.

  • Figure 1: a control is missing. Authors should also report the level of TNF-a and IL-6 of AGS and mouse treated with Capsaicin but non-infected. It is especially important as the authors appear to be using an extremely high dose of the drug (100M).

  • The non-infected control treated with Capsaicin is missing throughout the whole article. Authors revise the manuscript include the data throughout.

  • Figure 2A, B, C & D: What the “infected tissue” the authors are talking about? Infected cells or infected mouse? The figure legends need to clearly stipulate what is tested!

  • Figure 2E: Authors need a loading control such as B-actin, tubulin or GAPDH.

  • Figure 2E & 2F: The changes appear very small and the western blot pretty dirty. In addition, NF-KB is not constant. The authors said they performed the experiment in triplicate. Can they please provide the 3 sets of Western blot used for the quantification?

  • Does Capsaicin affect NF-KB nuclear translocation?

  • Western blots loading controls should be consistent throughout the manuscript. Why did the authors pick 2 different loading control in panels Figure 3D (B-Actin) and 4A (a-Tubulin). The rationale needs to be explained.

  • To prove Capsaicin reduces NF-KB binding to mir21 and miR223 promoters, authors should perform ChIP assay.

  • Figure 3C and 3D: Capsaicin is known to inhibit H pylori infection as stated by the authors lines 74 to 76. Therefore, CagA would naturally be decreased in these cells and thus there would be decrease inflammation. Therefore, Capsaicin may have absolutely no effect on NF-KB beyond the reduced infection. This is an important issue and needs to be addressed. Effect of Capsaicin treatment on cells either treated with CagA protein or expressing CagA protein should be investigated to show that the effect of Capsaicin is through the actuall NF-KB signaling pathway and do not just the result from lower infection.

  • Some figures such as Figure 2F or 3E are not called within the manuscript.

  • Figure 4: It looks like total Akt proteins is greatly decreasing with Capsaicin treatment.
    • Please provide quantification for total Akt levels normalized to loading control Tubulin.
    • How many times were these western blots repeated?
    • Please provide original data for the different independent repeats

Minor comments:

  • The bars between columns which present statistical significance are all over the place. Please keep them centered to their appropriate column.

  • Figure formatting is not homogenous. The letters attached to each panel varies in format between figures and need to be corrected.

  • Western blot presentation needs to be improved:
    • Western blots should all indicate some size reference (kDa),
    • Normalized band intensity should be written below the band.

  • Supplemental Figure 1 is useful and should be called during the results section (line 93 probably).

  • Line 134 GAPDH is not a loading control (that’s for Western Blot), it’s a reference gene.

  • Figure 2B: the y-axis title has an issue, please fix the “α”.

  • Supplemental Figure 1 needs a figure legend.

  • Line 144: “mir21” should be miR21.

  • Line 171: “Mir21 and mir223” should have a R capitalized.

  • Figure 5C quality might be a little low, would you have an higher resolution for that panel?

Author Response

Reviewer 1

Major comments:

  • Lines 74 to 76: Authors need reference(s) for their statement that Capsaicin is known to inhibit H pylori infection.
  • We acknowledge a thorough review of the manuscript. There has been a typing error in the statement. This will be ‘Capsaicin is known to inhibit inflammation during H pylori infection”It has been included in the manuscript. Reference(34) is also added.

  • Figure 1: Is this a typo or are the authors using 100M concentration of the drug? Please provide cell viability assay for that drug.
  • This is a typing error. It will be 100mM concentration. We have incorporated it in the manuscript. We have also performed cell viability assay and incorporated it in Supplementary  figure 1.

  • Figure 1: a control is missing. Authors should also report the level of TNF-a and IL-6 of AGS and mouse treated with Capsaicin but non-infected. It is especially important as the authors appear to be using an extremely high dose of the drug (100M).
  • The level of TNFa and IL6 with control (only capsaicin treated) has been incorporated in the figure. We have added non infected control treated with capsaicin for in vitro experiments only. It is not possible for in vivo experiments as the total time of capsaicin treatment in mice is 40 days.
  • The non-infected control treated with Capsaicin is missing throughout the whole article. Authors revise the manuscript include the data throughout.
  • We have added non infected control treated with capsaicin for in vitro experiments only. It is not possible for in vivo experiments as the total time of capsaicin treatment in mice is 40 days.

  • Figure 2A, B, C & D: What the “infected tissue” the authors are talking about? Infected cells or infected mouse? The figure legends need to clearly stipulate what is tested!
  • In Figure 2A,B,C,D infected tissue is about mice tissue samples infected with H pylori. (in vivo samples). This has been included in the figure legends.

  • Figure 2E: Authors need a loading control such as B-actin, tubulin or GAPDH.
  • We have incorporated tubulin in Fig.2E as loading control.

  • Figure 2E & 2F: The changes appear very small and the western blot pretty dirty. In addition, NF-KB is not constant. The authors said they performed the experiment in triplicate. Can they please provide the 3 sets of Western blot used for the quantification?
  • We have included the three sets of western blots used for quantification in the supplementary figure for Fig.2E and F.

  • Does Capsaicin affect NF-KB nuclear translocation?
  • Yes, Capsaicin inhibits nuclear translocation of NF-kB. It has been reported earlier(Ref 34). We have also performed promoter assay with nuclear extracts. Here we have observed inhibition of NF-kB binding to promoter elements due to capsaicin treatment in AGS cells.NF-kB can bind to promoter elements when it enters the nucleus during H pylori infection. Capsaicin inhibits nuclear translocation of NF-kB , hence promoter activity is less as compared to H pylori infected cells.
  • Western blots loading controls should be consistent throughout the manuscript. Why did the authors pick 2 different loading control in panels Figure 3D (B-Actin) and 4A (a-Tubulin). The rationale needs to be explained.
  • We have used b actin for in vitro experiment only. For in vivo experiments, we have used tubulin. For in vivo samples, b actin was not clear in western blots. Hence, we used tubulin throughout for in vivo samples.

  • To prove Capsaicin reduces NF-KB binding to mir21 and miR223 promoters, authors should perform ChIP assay.
  • To prove capsaicin reduces NF-KB binding to promoter elements we have performed promoter activity assay instead of Chip assay as we have at present the promoter assay kit. Figure 3C and 3D: Capsaicin is known to inhibit H pylori infection as stated by the authors lines 74 to 76. Therefore, CagA would naturally be decreased in these cells and thus there would be decrease inflammation. Therefore, Capsaicin may have absolutely no effect on NF-KB beyond the reduced infection. This is an important issue and needs to be addressed. Effect of Capsaicin treatment on cells either treated with CagA protein or expressing CagA protein should be investigated to show that the effect of Capsaicin is through the actuall NF-KB signaling pathway and do not just the result from lower infection.
  • There has been a typing error in the statement. This will be ‘Capsaicin is known to inhibit inflammation during H pylori infection in in vitro conditions” It has been incorporated in the manuscript. Reference is also added(34).   Here, we have shown that capsaicin reduces NF-kB promoter activity and affects transcriptional targets of NF-kB like IL6, TNFa, IL-1b and NLRP3. It also reduced miRNA expressions which are transcriptional targets of NF-kB. Hence, we have proved that capsaicin has effect on NF-kB apart from its antimicrobial activity. It reduces CagA expression as it reduces infection. There are several reports showing anti H pylori effect along with reduction of NF-kB activity. Example: silibinin, probiotics, aspirin  and other plant derived compounds showed reduced NF-kB activity and at the same time also reduced H pylori infection( Kyunghwa Cho1J Cancer Prev 26(2):118-127, June 30, 2021, Johnson-et al. Probiotics reduce bacterial colonization and gastric inflammation in H. pylori-infected mice. Dig Dis Sci 2004; 49, Sarkar et al, Curcumin as a potential therapeutic candidate for Helicobacter pylori associated diseases, World J Gastroenterol. 2016). Similarly, we have shown that Capsaicin reduces inflammation by inhibiting NF-kB activation and at the same time also reduces H pylori colonisation. Further investigations are not possible at present to show direct effect of capsaicin on CagA.
  • Some figures such as Figure 2F or 3E are not called within the manuscript.
  • These are arbitrary densitometry data of the western blots. We have incorporated within the manuscript.

  • Figure 4: It looks like total Akt proteins is greatly decreasing with Capsaicin treatment.
    • Please provide quantification for total Akt levels normalized to loading control Tubulin.
    • How many times were these western blots repeated?
    • Please provide original data for the different independent repeats
    • We have incorporated triplicate data for total akt and tubulin. We have also provided quantification data. Western blots were repeated thrice.(Supplementary)

Minor comments:

  • The bars between columns which present statistical significance are all over the place. Please keep them centered to their appropriate column.
  • We have corrected the figures.

  • Figure formatting is not homogenous. The letters attached to each panel varies in format between figures and need to be corrected.
  • Formatting has been corrected for all the figures.

  • Western blot presentation needs to be improved:
    • Western blots should all indicate some size reference (kDa),
    • Normalized band intensity should be written below the band.
    •  We have included all the points as suggested by Reviewer.
  • Supplemental Figure 1 is useful and should be called during the results section (line 93 probably).
  • As suggested we have incorporated Supplementary Figure 1 in Figure1 .

  • Line 134 GAPDH is not a loading control (that’s for Western Blot), it’s a reference gene.
  • In line 134 we have included gapdh as housekeeping gene.

  • Figure 2B: the y-axis title has an issue, please fix the “α”.
  • We have corrected the y axis in Figure 2B.

  • Supplemental Figure 1 needs a figure legend.
  • We have incorporated legend in Figure1.

  • Line 144: “mir21” should be miR21.
  • We have corrected line 144
  • Line 171: “Mir21 and mir223” should have a R capitalized..
  • We have used capital R in line 171

  • Figure 5C quality might be a little low, would you have an higher resolution for that panel?
  • For Figure 5C we have used higher resolution picture (Supplementary Figure 2).

Reviewer 2 Report

Major corrections: 

  • I believe that the Introduction should be improved. There are quite plenty of logical mistakes, together with repetitive sentences. Far too much attention was paid to the description of the pro-inflammatory cascades, while the description of capsaicin was practically completely omitted (only one sentence appears!). The description of capsaicin should definitely be expanded with information on the biochemistry of this compound and its use in medicine and other sectors, e.g. pharmaceuticals 
  • Please explain why 100 M of capsaicin was used as the only tested concentration? How does this relate to cytotoxicity against human cells? Why were no other concentrations used? Ideally, please use the literature data in your argument. 

Minor corrections: 

  • “but it causes other gastric complications like ulcer and gastritis” -> but it also causes other gastric complications like ulcers and gastritis [line 38-39] 
  • “with H pylori (SS1 strain) at 1:100 MOI for 4 h. This is a CagA positive strain of H pylori.” -> with H pylori (SS1 strain, a CagA-positive isolate) at 1:100 MOI for 4 h. [lines 88-89] 
  • “capsaicin in in vivo model of H pylori we infected mice with H pylori (SS1 strain)” -> capsaicin in in vivo model we infected mice with H pylori SS1 strain [line 92] 
  • “as compared to H pylori infection.” -> as compared to a H pylori-infected, non-treated group [line 94] 
  • “ELISA was performed with the media collected after infection from Control (CON)” -> Logical mistake, the term “infection” cannot be related to the control group without infection; please change [line 106] 
  • “capsaicin infected treated mice” -> the same as above (logical mistake), mice cannot be infected with capsaicin [line 110] 
  • “However, the role of capsaicin in inhibiting inflammation in vivo model is unknown. Here, we have shown that capsaicin is able to reduce NF- B activation in in vivo model.” -> However, the role of capsaicin in inhibiting inflammation in vivo model is unknown. Therefore, the main aim of the current study was to determine an ability of capsaicin to reduce NF-B activation in in vivo model. [lines 228-230] 
  • Section 4.5 -> Please change the font 
  • Sections 4.10 + 4.10.1 -> Please expand the description of the in silico research, because the information about the program used is definitely not enough and makes it impossible to reproduce the research. Please describe the parameters that were used for the calculations. 

Author Response

Reviewer 2

Major corrections: 

  • I believe that the Introduction should be improved. There are quite plenty of logical mistakes, together with repetitive sentences. Far too much attention was paid to the description of the pro-inflammatory cascades, while the description of capsaicin was practically completely omitted (only one sentence appears!). The description of capsaicin should definitely be expanded with information on the biochemistry of this compound and its use in medicine and other sectors, e.g. pharmaceuticals 
  • We have thoroughly revised the introduction according to Reviewer 1‘s suggestions. We have incorporated a detailed description of capsaicin in one paragraph. For description of proinflammatory cascades we have deleted the repetitive sentences.
  • Please explain why 100 M of capsaicin was used as the only tested concentration? How does this relate to cytotoxicity against human cells? Why were no other concentrations used? Ideally, please use the literature data in your argument. 
  • This is a typing error. It will be 100mM concentration We have incorporated it in the manuscript. We have also performed cell viability assay and incorporated it in the supplementary figure 1. We have started with 100 mM according to earlier report (ref 36 ).
  •  

Minor corrections: 

  • “but it causes other gastric complications like ulcer and gastritis” -> but it also causes other gastric complications like ulcers and gastritis [line 38-39] 
  • We have included the corrections.
  • “with H pylori (SS1 strain) at 1:100 MOI for 4 h. This is a CagA positive strain of H pylori.” -> with H pylori (SS1 strain, a CagA-positive isolate) at 1:100 MOI for 4 h. [lines 88-89] 
  • We have included the corrections.
  •  
  • “capsaicin in in vivo model of H pylori we infected mice with H pylori (SS1 strain)” -> capsaicin in in vivo model we infected mice with H pylori SS1 strain [line 92] 
  • We have included the corrections.
  •  
  • “as compared to H pylori infection.” -> as compared to a H pylori-infected, non-treated group [line 94] 
  • We have included the corrections.
  •  
  • “ELISA was performed with the media collected after infection from Control (CON)” -> Logical mistake, the term “infection” cannot be related to the control group without infection; please change [line 106] 
  • We have included the corrections.
  •  
  • “capsaicin infected treated mice” -> the same as above (logical mistake), mice cannot be infected with capsaicin [line 110] 
  • We have included the corrections.
  •  
  • “However, the role of capsaicin in inhibiting inflammation in vivo model is unknown. Here, we have shown that capsaicin is able to reduce NF- B activation in in vivo model.” -> However, the role of capsaicin in inhibiting inflammation in vivo model is unknown. Therefore, the main aim of the current study was to determine an ability of capsaicin to reduce NF-B activation in in vivo model. [lines 228-230] 
  • We have included the corrections.
  •  
  • Section 4.5 -> Please change the font 
  • We have included the corrections.
  •  
  • Sections 4.10 + 4.10.1 -> Please expand the description of the in silico research, because the information about the program used is definitely not enough and makes it impossible to reproduce the research. Please describe the parameters that were used for the calculations.
  • We have expanded the description of in silico research. We have described the parameters used for calculations (materials and methods). 

Round 2

Reviewer 1 Report

Authors made significant effort to address most of my comments. I still have a minor  issue with presentation: again specifically the bars to indicate significance are really not centered, please take care of those before publication as it looks too careless right now.

Author Response

Comments and Suggestions for Authors

  • Authors made significant effort to address most of my comments. I still have a minor issue with presentation: again specifically the bars to indicate significance are really not centered, please take care of those before publication as it looks too careless right now.

  • We have incorporated the corrections as suggested by Reviewer 1.

Reviewer 2 Report

Thank you very much for responding to my comments. I believe that at this point the quality of the manuscript has improved.

I am asking only for a slight change in the Fmage 1 in the Supplementary Materials (please change "µm" to "µM" both in the graphic and in the description below it).

Author Response

Comments and Suggestions for Authors

  • I am asking only for a slight change in the Fmage 1 in the Supplementary Materials (please change "µm" to "µM" both in the graphic and in the description below it).

  • We have incorporated the corrections in supplementary as suggested by Reviewer 2.
